# AgNO_3_ Sterilizes Grains of Barley (*Hordeum vulgare*) without Inhibiting Germination—A Necessary Tool for Plant–Microbiome Research

**DOI:** 10.3390/plants9030372

**Published:** 2020-03-17

**Authors:** Victoria Munkager, Mette Vestergård, Anders Priemé, Andreas Altenburger, Eva de Visser, Jesper Liengaard Johansen, Flemming Ekelund

**Affiliations:** 1Department of Biology, University of Copenhagen, 2100 Copenhagen, Denmarkjljohansen@bio.ku.dk (J.L.J.);; 2Department of Agroecology, Aarhus University, 4200 Slagelse, Denmark; 3The Arctic University Museum of Norway, UiT—The Arctic University of Norway, 9006 Tromsø, Norway

**Keywords:** Barley, *Hordeum vulgare*, seed sterilization, grain sterilization, AgNO_3_, silver nitrate, plant–microbiome, endophyte sterilization

## Abstract

To understand and manipulate the interactions between plants and microorganisms, sterile seeds are a necessity. The seed microbiome (inside and surface microorganisms) is unknown for most plant species and seed-borne microorganisms can persist and transfer to the seedling and rhizosphere, thereby obscuring the effects that purposely introduced microorganisms have on plants. This necessitates that these unidentified, seed-borne microorganisms are removed before seeds are used for studies on plant–microbiome interactions. Unfortunately, there is no single, standardized protocol for seed sterilization, hampering progress in experimental plant growth promotion and our study shows that commonly applied sterilization protocols for barley grains using H_2_O_2_, NaClO, and AgNO_3_ yielded insufficient sterilization. We therefore developed a sterilization protocol with AgNO_3_ by testing several concentrations of AgNO_3_ and added two additional steps: Soaking the grains in water before the sterilization and rinsing with salt water (1% (*w*/*w*) NaCl) after the sterilization. The most efficient sterilization protocol was to soak the grains, sterilize with 10% (*w*/*w*) AgNO_3_, and to rinse with salt water. By following those three steps, 97% of the grains had no culturable, viable microorganism after 21 days based on microscopic inspection. The protocol left small quantities of AgNO_3_ residue on the grain, maintained germination percentage similar to unsterilized grains, and plant biomass was unaltered. Hence, our protocol using AgNO_3_ can be used successfully for experiments on plant–microbiome interactions.

## 1. Introduction

Seeds are inhabited by a diverse array of viable microorganisms, which can affect subsequent plant development [1]. To identify the effects of specific, purposely introduced microorganisms on plant growth requires that unidentified seed-borne microorganisms are removed.

Seed-borne microorganisms are located on the seed surface (epiphytes) or inside the seed (endophytes) [2]. Endophytic bacteria have been found in the embryo and endosperm of seeds [3,4,5] and they belong primarily to the phyla Proteobacteria, Actinobacteria, Firmicutes, and to a lesser extend Bacteroidetes [1,6,7,8]. This dominance of the phyla Proteobacteria, Actinobacteria, and Firmicutes as grain endophytes, has also been shown specifically for barley [9,10]. Fungal endophytes mostly belong to either of the two classes Dothideomycetes and Tremellomycetes [2].

Several transmission routes for endophytic bacteria have been proposed. Endophytes can be transferred to seeds maternally through the vascular system or paternally via the gamete [5]. Viable bacteria transferred by spraying the flowers of parent plants were found inside the embryo of seeds [11], which led the authors to suggest that air or contact with insects could be a pathway for the introduction of seed endophytes. Berg and Raaijmakers [5] suggest that for wild plant species, passage through animal guts could affect the seed microbiome, while seed epiphytes in cultivated plants can be transferred during post-harvest storage [12]. Those factors can be plant-specific [5], which has also been confirmed specifically for barley. Two independent studies on grain endophytes of barley found variation in the relative abundance of bacteria across different barley cultivars [9,10]. Additionally, it was highlighted by the authors of one of the studies [9] that they found *Xanthomonas* to be the most abundant genus for one cultivar, whereas it was absent as a grain endophyte in the second study [10].

Consequently, it becomes evident that nothing is known about the microbiome of seeds of specific plant species, or even cultivars without analysis [13]. It has been shown that seed endophytes can affect germination, seedling growth, and pathogen resistance [14,15] and two separate studies found evidence, which suggests that seed-borne microorganisms can relocate to the seedling and rhizosphere microbiome [8,16]. In summary, the seed microbiome is unknown for most plants and it can affect the plant and rhizosphere microbiome composition, which necessitates the removal of the seed microbiome (inside and surface) before studying the interaction between plants and microorganisms in a meaningful way.

Studies with *Arabidopsis* can circumvent these problems by using sterile seeds, which are readily available. For most plants, however, seeds which are sterile or gnotobiotic (all organisms are identified) are not available, and researchers must rely on seed sterilization. Therefore, a rapid, reliable sterilization protocol is essential. Commonly applied chemical seed sterilization agents include sodium hypochlorite (*Arabidopsis thaliana* [17,18], *Triticum aestivum* [19], *Gossypium* [20], *Hordeum vulgare* [21]), chlorine gas (Cotton, [20]), mercuric chloride (*Triticum aestivum* [19], *Stevia rebaudiana* [22]), hydrogen peroxide (Cotton [20], *Hordeum vulgare* [23,24,25]), propylene oxide gas (*Triticum aestivum* [19]), and silver nitrate (*Triticum aestivum* [19], *Hordeum vulgare* [26,27]). Methods for physical seed sterilization include gamma irradiation (*Hordeum vulgare* [28]) and heat (*Hordeum vulgare* [29], various species [30]). However, for most of the chemical agents, there is no consensus on the best protocol regarding duration of exposure, concentration, and pre- and post-sterilization treatments, and it is evident that efficacy of a protocol can differ even between cultivars of the same species [20]. At present, no single protocol is the gold standard for seed sterilization and to our knowledge, no study has provided a method to sterilize seeds beyond the surface whilst also showing that the subsequent germination and plant growth was unaffected. This hampers any progress in plant-microbe interaction research. 

In this study, we aimed to find the most efficient protocol for grain sterilization for experiments on plant–microorganism interactions under controlled conditions. We used four criteria to evaluate the efficacy of the protocol: 1. High proportion of grains with no culturable, viable microorganism, 2. high germination percentage, 3. no residual sterilizing effect after completion of protocol, and 4. no effect of sterilization on subsequent plant growth. We initially tested three sterilizing agents (NaClO, H_2_O_2_, and AgNO_3_) and continued to optimize the protocol with the most promising agent, i.e., AgNO_3_. We then proceeded to test a range of AgNO_3_ concentrations along with pre- and post-sterilization treatments. We hypothesized that AgNO_3_ would kill both epiphytes and endophytes, as AgNO_3_ penetrated the seed coat of *Xanthium glabratum* to a similar extent as water during seed imbibition [31]. We hypothesized that soaking grains for 20 h as a pre-sterilization treatment would soften the grain surface and thereby make it more permeable to liquids and that the presence of water would activate dormant microorganisms and make them more susceptible to sterilization. Finally, we hypothesized that rinsing with dissolved NaCl after exposure to AgNO_3_ would stop the sterilization process via removal of Ag^+^ ions and by binding the Ag^+^ ions to Cl^−^ ions creating the solid AgCl which does not kill microorganisms. 

## 2. Results

The term sterility will henceforth be used to describe conditions where no microorganisms on or near the grain or seedling were identified with a dissecting microscope after 21 days. This method does not eliminate that viable, but nonculturable cells, microcolonies invisible with a dissecting microscope, or cells with inhibited growth, yet alive were present.

The results from Step 1 (Section 4.2), aimed at finding the best sterilization agent to proceed with, revealed that 9% (*w*/*w*) NaClO and 3% (*w*/*w*) H_2_O_2_ resulted in only 6.7% and 0% sterility respectively, thus rendering 1% (*w*/*w*) AgNO_3_, with 83.3% sterility (Figure 1) the most promising candidate for developing a protocol for sterilizing barley grains.

### 2.1. Germination was Unaffected by AgNO_3_ when Grains were Rinsed in NaCl

Germination percentage on day 21 remained high as AgNO_3_ concentration increased (Figure 2). Only in two treatment combinations (dry grains, rinsed in sterile ddH_2_O, on Potato Dextrose Agar (PDA) and soaked grains, rinsed in sterile ddH_2_O, on Tryptic Soy Agar (TSA)), did germination decline in response to increasing AgNO_3_ concentrations. For these two treatment combinations, germination was significantly reduced at 3% and 10% (*w*/*w*) AgNO_3_. Grains on TSA plates with cycloheximide displayed inhibited germination compared to grains on PDA plates for all treatment combinations. On PDA plates, there were no significant differences between the four combinations of pre- and post-sterilization treatments at AgNO_3_ concentrations 0%–1% (*w*/*w*). At higher concentrations (3%–10% (*w*/*w*)), the grains rinsed in ddH_2_O had fewer germinated seeds than the grains that were rinsed in NaCl.

Control seeds (without AgNO_3_) had no significant difference in germination between dry and soaked grains after 21 days of incubation on PDA plates (Appendix A). On TSA plates, soaking led to an increase in germination. These results are consistent with the results when grains were exposed to AgNO_3_ and similar to results on day 7 (Appendix A). 

Comparing the germination percentage on PDA plates on day 7 and day 21, no significant difference was found for grains rinsed in NaCl after sterilization regardless of AgNO_3_ concentration and pre-treatment (dry vs soaked) (Appendix A and Appendix A). Contrastingly, grains rinsed only in sterile ddH_2_O did have a significantly lower germination percentage on day 7 compared to day 21 (Appendix A and Appendix A). This difference between day 7 and 21 was significant for dry grains regardless of AgNO_3_ concentration except for the control concentration (0% (*w*/*w*) AgNO_3_). For soaked grains the difference in germination between day 7 and 21 only occurred at the three highest AgNO_3_ concentrations (1%–10% (*w*/*w*)). At lower AgNO_3_ concentrations (0%–0.2% (*w*/*w*)) there were no discernible differences.

In summary, the number of germinated grains was unaffected by AgNO_3_, regardless of concentration, and germination was not delayed as long as grains were rinsed in NaCl. In contrast, when grains were rinsed only in sterile ddH_2_O, germination was delayed and high concentrations of AgNO_3_ reduced the number of germinated grains. 

### 2.2. AgNO_3_ Effectively Killed Microorganisms

AgNO_3_ was found to be a very effective sterilizing agent. On PDA medium, which unlike TSA contained no fungicide, sterility was as high as 98% for soaked grains, sterilized with 10% (*w*/*w*) AgNO_3_ and rinsed in sterile ddH_2_O (Appendix A and Appendix A). For germinated grains, >99% of the grains were 100% sterile (Figure 3, Appendix A). The efficacy of AgNO_3_ depended on concentration. Sterility of germinated grains increased with AgNO_3_ concentration until it reached a level where a higher concentration of AgNO_3_ no longer improved sterility. The different combinations of pre- and post-sterilization treatments reached this level at different concentrations of AgNO_3_. The treatment combination with dry grains rinsed in sterile ddH_2_O reached high sterility at the lowest concentration of 1% (*w*/*w*) AgNO_3_. Soaked grains rinsed in NaCl reached maximum sterility at the highest concentration of 10% (*w*/*w*). Dry grains rinsed in NaCl were not considered as the method had limited effect (Figure 3 and Appendix A). The trend was similar when looking at sterility of both germinated and ungerminated grains (Appendix A and Appendix A). 

### 2.3. Pre-Treatment Soaking for 20 Hours Increased Sterility and Post-Sterilization Treatment NaCl Halted the Sterilization Process of Ag^+^


Dry grains, rinsed in NaCl displayed poor sterilization regardless of AgNO_3_ concentrations, on both PDA and TSA, with maximum 6% and 20% of grains being 100%, sterile respectively. Soaking grains prior to sterilization significantly increased sterility when grains were rinsed in NaCl afterwards. Soaking did not affect sterility on PDA plates when grains were rinsed only in ddH_2_O (P >.05). On PDA plates, we observed fewer contaminated grains after AgNO_3_ treatment when grains were rinsed only in ddH_2_O compared to rinsing with NaCl, except at 10% (*w*/*w*) AgNO_3_. These differences in sterility between pre- and post-sterilization treatments were similar for the observed sterility of germinated grains (Appendix A). 

### 2.4. Quantity of Sterile Grains Cannot be Assesed after 7 Days as Some Contamination Occurs Later

A comparison of the number of grains with 100% sterility between day 7 and 21 showed that regardless of AgNO_3_ concentration, fewer grains were 100% sterile after 21 days than after 7 (Appendix A and Appendix A). This finding holds true for dry grains, soaked grains, grains rinsed in sterile ddH_2_O and for grains rinsed in NaCl incubated on both PDA and TSA. The control treatments (0% (*w*/*w*) AgNO_3_) were one of the three exceptions to this finding as no grains were 100% sterile at either time points. The two other exceptions were treatments 10% (*w*/*w*) AgNO_3_, soaked grains, sterile ddH_2_O on PDA and 10% (*w*/*w*) AgNO_3_, dry grains, sterile ddH_2_O on TSA where no difference between time points was found. 

### 2.5. AgNO_3_ Left Less Residual Ag on Grains When Rinsed in NaCl

In Step 3, we investigated whether residual Ag was left on the grains after sterilization in AgNO_3_. All treatment combinations, except 1% (*w*/*w*) AgNO_3_, soaked grains rinsed in NaCl, resulted in significantly more residual Ag on grains than the control. The quantity of residual Ag for non-control AgNO_3_ concentrations ranged from 5.39–588 µg Ag per g grain, which is an up to 375-fold increase compared to the control. 

When grains were rinsed in 1% (*w*/*w*) NaCl after sterilization, residual Ag in grains did not increase significantly with increasing concentration of AgNO_3_ (Figure 4). Likewise, when grains were rinsed in 1% (*w*/*w*) NaCl after sterilization, the residual Ag in grains was not different between grains that were dry and soaked. The trend looked different when grains were rinsed in sterile ddH_2_O after sterilization. In that case, the quantity of residual Ag on grains did depend on the pre-sterilization treatment and concentration of AgNO_3_. Dry grains retained more residual Ag than grains soaked for 20 h when rinsed only in sterile ddH_2_O and sterilized in 3%–10% (*w*/*w*) AgNO_3_. At 1% (*w*/*w*) AgNO_3_, dry grains had 175% more residual Ag than soaked grains, albeit this difference was not statistically significant due to high variance. At AgNO_3_ concentrations below 1% (*w*/*w*), the quantity of residual Ag was not discernibly different between any of the treatment combinations when compared within the same concentration of AgNO_3_. 

### 2.6. The Harshest AgNO_3_ Protocol Led to Delayed Germination but Biomass was Unaffected

Dry grains sterilized with 3% (*w*/*w*) AgNO_3_ and rinsed only in sterile ddH_2_O showed a decline in number of germinated grains, a delay in germination time, and the highest quantity of residual Ag. In Step 4, grains were sterilized by this treatment combination and subsequent plant growth was monitored to identify whether AgNO_3_ negatively affected plant growth when exposed to the most severe AgNO_3_ protocol. Plants from grains that had not been subjected to AgNO_3_ emerged three days earlier than the sterilized group (Figure 5A). In contrast, we observed no difference between dry weight of shoots of the untreated and the sterilized treatments (Figure 5B). Root weight showed the same pattern as shoot weight (Appendix A). 

## 3. Discussion

In this paper, we describe a method that leads to 100% sterility in almost all grains (98% ± 2) using AgNO_3_. This effect could even be maintained for 21 days of plant growth under conditions favoring microbial growth (PDA and TSA media). 

A successful protocol for grain sterilization should produce grains which are 100% sterile without affecting subsequent plant growth. Additionally, it is essential that the antimicrobial effect halts once the biocide has served its purpose. This is especially true if grains are to subsequently be used for experiments involving microorganisms, e.g., for manipulation of the grain, plant or rhizosphere microbiome. 

Rinsing grains with NaCl aided the removal of Ag since the Ag^+^ ions readily react with Cl^−^ ions to create the crystalline solid AgCl [32]. The solid AgCl does not have antimicrobial qualities. Precipitation in our NaCl rinsing solutions was visible, thus significantly smaller quantities of residual Ag remained when grains were rinsed in NaCl compared to just water. This corresponded to a 42-fold decrease in Ag at the highest concentration of AgNO_3_ for dry grains. Contrary to grains rinsed only in water, the quantity of residual Ag remained consistent across all concentrations of AgNO_3_ despite a 200-fold increase for grains rinsed in NaCl, which indicates that the NaCl solutions provided sufficient availability of Cl^−^. 

The percentage of grains that were 100% sterile, when dry grains were rinsed in NaCl was low (1%–6%), regardless of AgNO_3_ concentration, an efficacy comparable to that of grains exposed to water without any AgNO_3_ (0% sterility). This cannot be explained by AgNO_3_ lacking antimicrobial potential as 92% of the dry grains rinsed in water instead of NaCl had no bacteria or fungi. However, it seems to support our hypothesis that NaCl ensured that Ag^+^ ions seized to sterilize once rinsing commenced as has been reported previously [19,32], despite grains having more residual Ag left than controls. It is possible that at such low concentrations, Ag no longer had an effect or that all Ag^+^ ions were bound in the crystalline solid AgCl, thereby deactivating the sterilizing effect of Ag^+^. 

Residual Ag was similar for soaked and dry grains when rinsed in NaCl. Those low quantities of residual Ag were ineffective at sterilizing once rinsing had occurred as argued above and consequently the high sterility of soaked grains, rinsed in NaCl compared to the almost non-existing sterility of dry grains must be attributed to the pre-sterilization treatment, i.e., soaking, and not due to a lingering effect of the small quantity of residual Ag. While we can conclude that the quantity of residual Ag left on grains rinsed in NaCl had no effect on number of sterile grains, we cannot completely exclude the possibility, that the little residual quantity could affect growth and functioning of microorganisms. We therefore greatly encourage further studies that can elucidate whether such quantities of Ag after exposure to Cl^−^ ions have any effect on microorganisms as it thereby could affect studies where new microorganisms are introduced to the system. 

We hypothesized that soaking grains in water would enhance the antimicrobial effect of AgNO_3_, which we confirmed was true for grains rinsed in NaCl post sterilization (Figure 3). There are two possible theories, pertaining to imbibition, which could explain this. Imbibition is a strictly physical process, where the grain takes up water to start germination. Consequently, the hard grain surface softens [33], respiration and metabolic activity commence [34], and exudates, including carbon and nutrients, leak from the grain through ruptures in the surface [14]. At the time of AgNO_3_ exposure, soaked grains would already have initiated imbibition [33]. For dry grains, imbibition would have initiated during the 15 minutes of submersion in AgNO_3_, but the uptake of liquids, such as ddH_2_O with Ag^+^ ions, would still be low this early in the process [34]. During imbibition, the penetration of AgNO_3_ through the seed coat of *Xanthium glabratum* equaled that of water [31] demonstrating its potential to reach endophytic microorganism along with epiphytes. The increased permeability due to imbibition can therefore be a possible theory for why soaking would promote the antimicrobial effect of AgNO_3_. 

AgNO_3_ is biocidal due to the antimicrobial properties of free Ag^+^ ions, whereas pure, solid silver and AgCl are not antimicrobial [32]. AgNO_3_ is categorized as a disinfectant [32], meaning it is a broad-range biocide that, when the concentration is sufficiently high, kills bacteria, fungi, viruses, and protozoa through disruption of several mechanisms in contrast to only one specific mechanism [32,35]. Impairment of the respiratory chain [32,36] is one of the metabolic processes where proliferation is inhibited [35]. Secondly, in the bacterial cell, Ag^+^ can react with thiol groups of structural and enzymatic proteins thereby inactivating trans-membranous energy metabolism and electrolyte transport [32,35]. Thirdly, Ag^+^ can prevent replication of DNA [32,37]. As the antimicrobial effect of AgNO_3_ is associated with metabolic processes, it explains why AgNO_3_ is more effective on active microorganism and affect dormant microorganism less [32].

A common strategy to circumvent the problem of tolerance due to dormancy is to provide conditions that push the organisms into a metabolically active stage [38]. Microbial proliferation increases and metabolic processes are activated during and after imbibition due to the increased nutrient and water availability [14,39]. The second possible theory for the enhanced antimicrobial effect of AgNO_3_ due to soaking is that during soaking, dormant microorganisms become more active and subsequently more susceptible to stress. Both pathways for the efficiency-inducing effect of moisture have been proposed earlier to explain the increased antimicrobial effect of ethylene and propylene oxide when relative humidity of barley grains increased [40].

An optimal sterilization protocol would kill all microorganisms on and inside the grain without any adverse impact on the grain and its subsequent development. To check for any impairment in response to AgNO_3_, we evaluated germination percentage, which was defined as seedlings with a shoot longer than 5 cm and roots. When rinsed in NaCl, germination remained high regardless of increases in AgNO_3_ concentrations. In fact, germination was 92% and 96% at 10% (*w*/*w*) AgNO_3_ for dry and soaked grains, respectively compared to control treatments only rinsed in water where germination was 95% and 92%, respectively for dry and soaked grains. By contrast, dry grains rinsed only in water showed a decrease in germination percentage. We observed similar trends in germination difference between day 7 and 21. Grains rinsed in NaCl did not differ in germination percentage between the two points in time, whereas grains rinsed only in sterile ddH_2_O showed a delayed germination response. Our plant growth experiment, where grains were exposed to the harshest sterilization, similarly showed a delay in germination (3 days). The delay in development due to AgNO_3_ has previously been reported [41,42] alongside reports of reduced biomass and epigenetic changes [43,44]. It can be difficult to distinguish between the direct sterilization effects and indirect effects caused by the absence of endophytes, as they may also affect plant development [11,13,14]. At least some of the epigenetic changes due to Ag^+^ were associated with plant responses to microorganism [43], and these studies did not consider the neutralizing effect of Cl^−^ ions. Since NaCl eliminated the negative effect of AgNO_3_ on germination and we saw no reduction in plant biomass, it seems that the inhibitive impact of AgNO_3_ on plant development is not permanent but rather a transient inhibition of plants, which persists only as long as the sterilizing strength of Ag^+^ persists. Biomass accumulation was unaffected by sterilization with the harshest combination of pre- and post-sterilization treatments and concentration of AgNO_3_, when accounting for the 3-day delay in germination. This is another indicator that AgNO_3_ is an effective sterilizing agent that does not impact plant biomass negatively—as long as NaCl is used as a post-sterilization treatment. We cannot exclude the possibility that under certain conditions, Ag^+^ could affect plants in ways we have not investigated. A safety precaution for experiments investigating plant mechanisms, which we have not covered, could be to expose all treatments to the same sterilization protocol. 

In general, it appears a challenge to find a sterilization agent that kills all microorganisms but leaves the grain unharmed. The other sterilization agents commonly used (sodium hypochlorite, chlorine gas, mercuric chloride, hydrogen peroxide, propylene oxide gas, silver nitrate, and gamma irradiation) all led to altered seed germination [19,20,26,28]. Further, previous reports do not reach beyond 10 days. Yet, our data showed that sterility, measured as complete absence of bacteria and fungi on the surface of the plant (grain, shoot, root) or on the growth media (PDA or TSA), significantly declined during the time passed between day 7 and 21. Thus, more time than 7 days is needed to detect and identify all endophytic grain microorganisms, which may move from the internal tissue of the grain or seedling to colonize elsewhere. Seed endophytes have the ability to relocate to the soil and become significant players in the rhizosphere microbiome [8,16]. Microorganisms mostly thrive on the root and radicle as opposed to areas of the seed with less plant development [14]. This suggests that at least some microorganisms need the plant to thrive and could explain why some microorganisms need more time before initiating growth outside the seed. Hence, at least 21 days are needed to confidently establish whether a sterilization protocol has worked. We cannot exclude the possibility that the percentage of sterile grains had decreased if the experiment ran longer than 21 days. Slow-growing microorganisms or microorganisms which required different growing conditions than we provided (e.g., soil), could have been present but gone undetected, as we used a cultivation dependent method. Hence, we cannot eliminate the possibility that some Ag^+^ resistant microorganisms remained, which under different growth conditions would be viable. It is worth noting, that the antimicrobial effect of Ag^+^ ions is bacteriostatic in low concentrations but in high concentration it is bactericidal since the disruption of the trans- membranous energy metabolism is irreversible [35]. Ag^+^ has the ability to reach the inner compartments and tissue of seeds [31], and we recommend sterilizing with a very high concentration of AgNO_3_. Thus, the possibility of there being remnant, viable microorganisms in the grains, would require them to be resistant to Ag^+^. Resistance to Ag^+^ has been reported albeit for a limited number of microorganisms [35]. Yet, we encourage future studies that further our knowledge by employing cultivation-independent assessment of the sterility of grains post-sterilization, as we believe grain sterility is an essential tool to further our understanding of the interaction between plants and microorganism.

A common sterilization practice without variation in agent, exposure time, concentration and pre- and post-sterilization treatments could increase our knowledge on how seed-borne microorganisms affect plant–soil–microbiome interactions. Here, we propose that AgNO_3_ is an efficient sterilizing agent, with increased effects when grains are pre-soaked for 20 h followed by sterilization in 10% (*w*/*w*) AgNO_3_ and rinsing with 1% (*w*/*w*) NaCl after each of the three rinses in sterile ddH_2_O. This protocol results in 100% sterility for 97% of the grains, maintains a high germination rate (96%), and 98% of germinated grains were 100% sterile after 21 days of incubation while the quantity of residual Ag on grains remained low (12.79 ± 3.82 µg Ag per g grain). This method showed no delay in germination between 7 and 21 days as the maximum germination potential was reached by day 7, and plant biomass was not affected. 

## 4. Materials and Methods 

We conducted the evaluation of AgNO_3_ as an agent to sterilize grains without harming the subsequent plant growth in four steps. See Figure 6 for illustration of timeline and Section 4.2, Section 4.3, Section 4.4 and Section 4.5 for detailed information pertaining to the four steps. 

### 4.1. Grain Acquisition and Selection

We used undressed spring barley grains (*Hordeum vulgare* cv. Evergreen, Nordic Seed Galten, Denmark), a standard agricultural product, which likely had been exposed to a wide variety of soil microorganisms in the field and during storage, but had not been exposed to any sterilization against microbial contamination. Only undamaged grains within the weight range of 55 ± 5 mg were used.

### 4.2. Selection of Sterilization Agent

Initially we tested the efficiency of three different sterilization agents (NaClO, H_2_O_2_, and AgNO_3_). We tested NaClO as suggested by Speakman and Kruger [19]. Barley grains were soaked in tap water for 20 h, exposed to 9% (*w*/*w*) NaClO for 30 min, and washed three times in sterile ddH_2_O for 5 min. To test H_2_O_2,_ we followed Barampuram, Allen and Krasnyanski [20]. The grains were soaked in 1% (*w*/*w*) soap water (Vel Ultra, Colgate-Palmolive Company, Virum, Denmark) for 20 min, washed three times in sterile ddH_2_O for 5 min, exposed to 3% (*w*/*w*) H_2_O_2_ for 7 h, and washed three times in sterile ddH_2_O for 5 min. For AgNO_3_, grains were exposed to 1% (*w*/*w*) for 5 min and washed three times in sterile ddH_2_O for 5 min [25]. As controls to the three sterilization agents, a treatment consisted of soaking grains in tap water for 7 h followed by three rinses in sterile ddH_2_O for 5 min and a treatment where nothing was done to the grains prior to incubation. 

For each the 5 different treatments, 30 seeds were exposed and then put on plates containing PDA prepared from 39 g of PDA) powder (Sigma) per L ddH_2_O. Each plate had 10 seeds, i.e., each treatment consisted of 3 plates. We checked for sterility after 6 days of incubation in the dark at 15°C as described in Section 4.3. 

### 4.3. Effect of AgNO_3_ on Sterility and Germination

To find the optimal protocol for sterilizing grains with AgNO_3_, we tested 6 different concentrations of AgNO_3_ in ddH_2_O: 0%, 0.05%, 0.2%, 1.0%, 3.0%, and 10.0% (*w*/*w*) combined with pre- and post-sterilization treatments. For each treatment combination, 120 barley grains were shaken at 150 rpm in 50 mL AgNO_3_ solution for 15 min. Prior to treatment with AgNO_3_, half of the grains were soaked in water for 20 h before AgNO_3_ exposure. The other half received no pre-sterilization treatment. After exposure to AgNO_3_, grains were rinsed three times for 5 min in sterile ddH_2_O on a rotary shaker. As a post-sterilization treatment, half of the grains were rinsed in a sterile 1% (*w*/*w*) NaCl solution for 5 min after each rinse in sterile ddH_2_O. See Table 1 for schematic overview of the experimental setup. 

After sterilization, grains were incubated in the dark at 15°C on two different growth media in petri dishes sealed with parafilm. To stimulate bacterial growth, we used TSA: 0.3 g Tryptic Soy Broth powder (Becton Dickinson, Franklin Lakes, New Jersey, USA), 15 g agar (Merck, Darmstadt, Germany) and 100 μg mL^−1^ cycloheximide (Sigma, St. Louis, USA) per L ddH_2_O. To stimulate fungal growth, we used PDA prepared from 39 g of PDA powder (Sigma) per L ddH_2_O. For each of the 22 treatment combinations, we set up 10 petri dishes of TSA and 10 of PDA, each with 5 grains, resulting in 50 replicate grains per medium, nested within petri dish.

Grains were scored for sterility and germination after 7 and 21 days. We also noted root and shoot development parameters (presence of shoot and root, length of shoot and number of roots) for each individual grain. We considered grains germinated if a shoot longer than 5 cm had developed and roots were present after 21 days. Less development than that was scored as ungerminated as seedling development should have progressed further than that in a 21-day timespan. 

We used a dissecting microscope (Olympus SZX16) to visually detect and count any fungal or bacterial presence on the surface of the grain, shoot, and root as well as on the growth media where the plant was located. When needed, suspected microorganisms were transferred to a slide and examined under a light microscope (Olympus BX50 at magnifications 100×–1000×). 

We converted the count data into a binary variable of absence/presence. A grain was only scored as sterile when bacteria and fungi on the grain, plant and in its proximity in the petri dish were completely absent. The score on day 7 helped to monitor the progress, but results for choosing the optimal sterilization protocol were based on the final score at day 21, as the objective was to not only surface sterilize but also to remove all endophytes capable of relocating to the exterior of the plant and its surroundings. 

### 4.4. Residues of Ag in and on Grains

Grains were treated as in step 2 (Section 4.3) with 40 grains in each treatment combination. After treatment, we measured Ag concentrations on three replicates of 0.5 g of grains dried to constant weight (ca. 10 grains). Grains were added to MARSXpress Teflon vessels (CEM, North Carolina, USA) and suspended in 10 mL of 32.5% (*v*/*v*) HNO_3_. The material was digested in a MARS 6 microwave (CEM, North Carolina, USA) using the plant material program (power: 1030–1800, ramp-time: 20–25 min, hold-time: 10 min, temperature: 200 °C). Ag content was measured with Atomic Absorptions Spectroscopy (PinAAcle 900T, PerkinElmer, Massachusetts, USA) with the ethylene flame at 328.1 nm.

### 4.5. Growth of Plants from Sterilized Grains 

To assess growth of plants from sterilized grains receiving the harshest sterilization protocol (dry seeds, 3% (*w*/*w*) AgNO_3_, rinsed in sterile ddH_2_O) without microbial contamination, we prepared microcosms in Falcon tubes (50 mL, Greiner Bio-One cat. No. 227 261) where the lids had been modified with two holes. Here, short plastic pipes were inserted (diameter: 7 mm and 4.5 mm)—the larger one for the plant and the smaller one for water supply. An artificial growth medium comprised of ignited (550 °C for 6 h) sand (quartz sand type: no. 2, particle size: 0.71–1.22 mm, Dansand A/S) and vermiculite in a 1:1 (v/v) mixture. All microcosm components were autoclaved and all work was done under sterile conditions. 

We germinated 150 unsterilized grains and 150 grains sterilized in 3% (*w*/*w*) AgNO_3_ and rinsed in sterile ddH_2_O as described in Section 4.3, in the dark at 15°C on PDA medium. The results from Steps 2 and 3 (Section 4.2, Section 4.3 and Section 4.4) showed that grains receiving 3% (*w*/*w*) AgNO_3_ without pre-soaking and post-sterilization treatment with NaCl had delayed germination, reduced number of germinated grains and the highest concentration of Ag residues. This concentration was therefore chosen for Step 4 where seedling development was monitored in response to AgNO_3_ sterilization. 

We subsequently placed 60 germinated, sterilized and 60 germinated, unsterilized grains individually in the Falcon tubes, and both plastic pipes in the lid were sealed with sterilized cotton. The cotton wool in the larger tube was reorganized around the growing seedling when needed, to keep the microcosms sealed without hindering plant growth. Seedlings that never emerged above the plastic tube were discarded. 

The Falcon tubes were placed in a growth chamber (14/10-hour light/dark conditions at 18°C). A solution containing all the essential nutrients was prepared by following a previously described recipe [45] although 600 µM KHCO_3_ and 300 µM NH_4_KNO_3_ replaced the 600 µM KNO_3_. The plants were checked every second to third day and the quantity of nutrient solution added was based on an estimated relative growth rate of 0.09 day^−1^ calculated from earlier experiments and the assumption that the nitrogen requirement equals 5% of the plant dry weight. The nutrient solutions were autoclaved before use. Sterile ddH_2_O was added to maintain a constant water supply of 15 mL. 

The date of seedling emergence above the plastic pipe was recorded and plants were harvested 25 days after that date. Shoots and roots were dried to constant weight at 50 °C and dry weight was recorded. 

### 4.6. Statistics 

For step 1, selection of sterilization agent (Figure 6), we tested differences in sterility and germination between treatments using a likelihood-ratio test on the main effect of treatment followed by a Tukey’s test on least-squares means. 

For step 2, the effects of AgNO_3_ on sterility and germination (Figure 6), we analyzed germination, sterility of grains (including germinated and ungerminated grains), and sterility of germinated grains using three separate generalized linear mixed models (binomially distributed). Here, pre-sterilization treatment, post-sterilization treatment and growth medium were fixed, categorical, binary variables and AgNO_3_ concentration a fixed, continuous variable. For all three models, we transformed the variable AgNO_3_ concentration using both a linear and quadratic term (“germination”: sqrt(AgNO_3_) + AgNO_3_, “sterility” and “sterility of germinated grans”: log(AgNO_3_) + AgNO_3_^2^). We did this to optimize the fit of the models and to allow for non-monotonic relations with AgNO_3_. We removed the 0% (*w*/*w*) AgNO_3_ concentration from the two models for sterility. No grains were 100% sterile at this concentration, resulting in perfect separation, and the models included log transformations. 

Since five grains were nested within the same petri dish, the normality of the random effect of petri dish was validated for all three response variables. All model diagnostics, likelihood-ratio tests, least-squares means and Tukey’s tests were run on the saturated models with interactions. The goodness of fit of the three models was validated separately based on the cumulative sum of residuals using the R package “gof” [46]. The used significance threshold for P-values for the “Kolmogorov–Smirnov” and “Cramer von Mises” tests was .05 based on 10,000 simulations. The model of germinated grains was fully validated, and the models of sterility and sterility of germinated grains were validated without serious deviations; though some P-values were below .05. We re-ran the cumulative sum-of-residuals tests separately for the four combinations of pre- and post-sterilization treatment. All models were then validated except for the treatment combination dry grains, rinsed in NaCl. This treatment had minimal effect on sterility and we therefore considered the models valid for our analysis. The difference in sterility and germination between day 7 and day 21 was analyzed separately for each combination of medium, and pre- and post-sterilization treatment, where the AgNO_3_ variable was transformed when needed to fit a model that was validated by its cumulative sum of residuals as describe above. 

For sterility, some concentrations of AgNO_3_ had perfect separation where observations were either all 1 or all 0 for both day 7 and day 21 making regression modelling obsolete. We excluded these data points from analysis and considered them to not be different. At other concentrations of AgNO_3_, only one time point had perfect separation whereas the other had variation. The data without perfect separation was then fitted as described above and the data with perfect separation was fitted as a regression model with the Firth’s bias reduction method (R package “logistf”) and AgNO_3_ as a factor variable. Whenever the modelling was split in two, it is indicated in the associated figures (Appendix A, and S3). 

For Step 3, residues of Ag in and on grains (Figure 6), the concentration of Ag left on grains was analyzed with a three-way ANOVA with pre- and post-sterilization treatments as fixed, categorical, binary variables. AgNO_3_ concentration was run as a fixed, categorical variable with six concentrations (0, 0.05, 0.2, 1, 3, and 10). It was not possible to validate a fitting model with AgNO_3_ as a continuous variable. The response variable, concentration of Ag, was log transformed. The model was run with a significant three-way interaction.

For all data from Step 2–3, a likelihood-ratio test was run to test for highest order interactions. Least-squares means (lsmeans) were used to estimate back transformed, predicted probabilities (± SE). The Tukey’s test on least-squares means was used to find differences between treatment combinations with adjustment for multiple comparisons. 

For Step 4, growth of plants from sterilized grains, a t-test was used, after validating with the Bartlett test that the two independent groups had equal variances, for difference between unsterilized and sterilized grains in i) dry weight and ii) days between transplanting and seedling emergence. 

All tests were based on a confidence level of 0.95 and significance threshold of .05. All analyses were run in R (version 3.3.1 3 (2016-06-21)) using the packages “lme4”, “gof”, “logistf”, and “lsmeans”.

## Figures and Tables

**Figure 1 plants-09-00372-f001:**
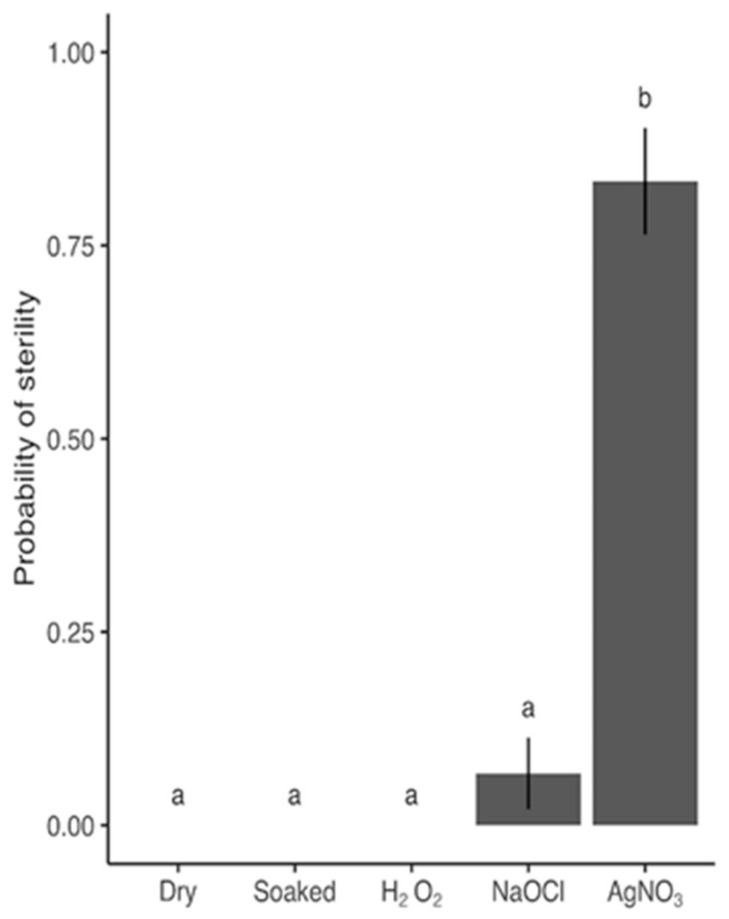
Probability of grains being 100% sterileafter 6 days in response to different sterilization agents (dry grains, grains soaked for 20 h, and grains sterilized with either 3% (w/w) H_2_O_2_, 9% (w/w) NaClO, or 1% (w/w) AgNO_3_). Bars and error bars are mean probability and SE of observed values (no. observations = 30). P-value from likelihood-ratio test of sterilization agents: <.0001. Shared letters between treatments denote that no statistically significant difference was found based on a Tukey’s test of least-squares means adjusted for multiple comparisons (5 estimates). Means and SE are specified in detail in Appendix A.

**Figure 2 plants-09-00372-f002:**
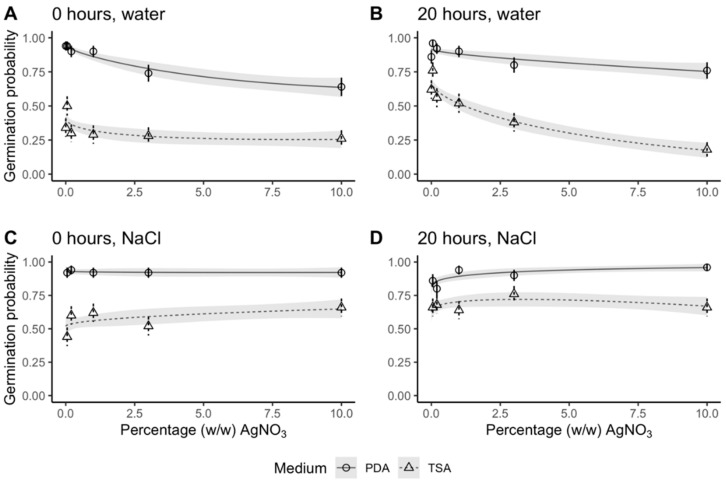
Germination probability of barley after 21 days in response to concentration of AgNO_3_. Symbols and error bars are mean probability and SE of observed values. Lines and shaded area are regression lines with SE of the generalized linear regression model (no. observations = 50 except 1% (*w*/*w*) AgNO_3_, dry grains rinsed in ddH_2_O on Tryptic Soy Agar (TSA) plates = 45). Treatments: (**A**) grains were dry before sterilization and rinsed in ddH_2_O after sterilization, (**B**) grains were soaked for 20 h before sterilization and rinsed in ddH_2_O after sterilization, (**C**) grains were dry before sterilization and rinsed in ddH_2_O and 1% (*w*/*w*) NaCl after sterilization, and (**D**) grains were soaked for 20 h before sterilization and rinsed in ddH_2_O and 1% (*w*/*w*) NaCl after sterilization. Data in circles and solid lines (both error bars and regression lines) are from grains germinated on Potato Dextrose Agar (PDA) and data in triangles and dashed lines (both error bars and regression lines) are from grains germinated on TSA. Least-squares means and Tukey’s test results are specified in detail in Appendix A.

**Figure 3 plants-09-00372-f003:**
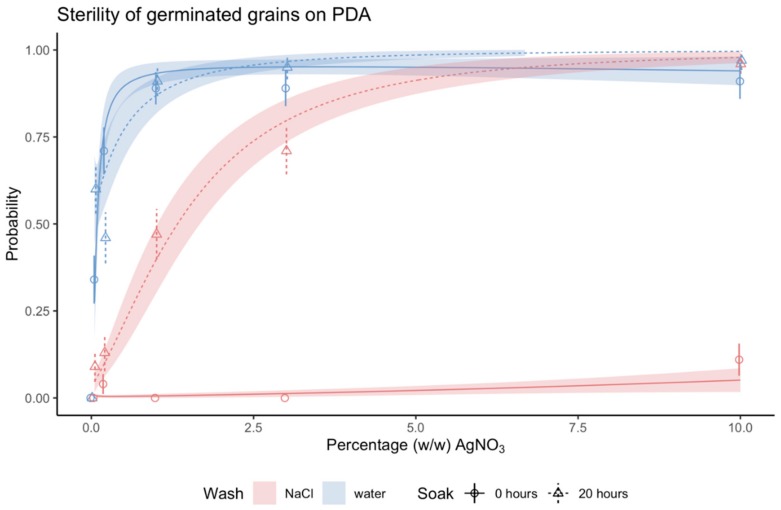
Sterility of germinated grains of barley after 21 days in response to concentration of AgNO_3_. Symbols and error bars are mean probability and SE of observed values. Lines and shaded area are regression lines with SE of the generalized linear regression model (no. observations = 32–48). Pre-sterilization treatment: Data in circles and solid lines (both error bars and regression lines) are from dry grains and data in triangles and dashed lines (both error bars and regression lines) are from grains soaked for 20 h prior to sterilization. Post-sterilization treatment: Data in blue are from grains rinsed in sterile ddH_2_O and data in red are from grains rinsed in 1% (*w*/*w*) NaCl. The four-way interaction of AgNO_3_ concentration, medium, pre-sterilization treatment, and post-sterilization treatment was statistically significant: P-value: .001. Least-squares means and Tukey’s test results are specified in detail in Appendix A.

**Figure 4 plants-09-00372-f004:**
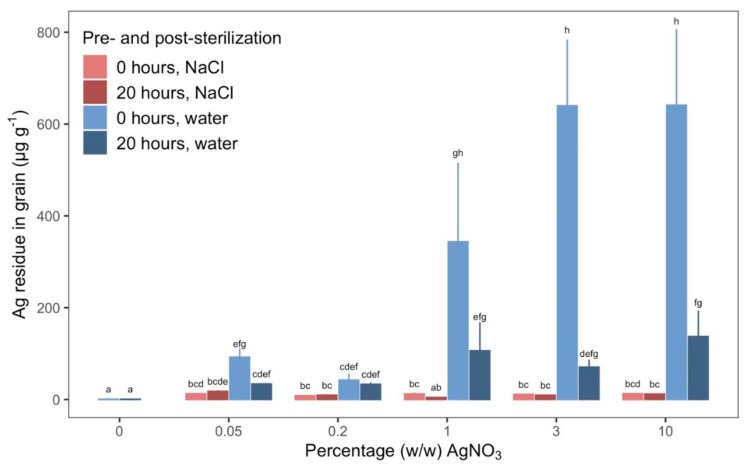
Residual Ag left on grains of barley from sterilization. Measurements were done after sterilization with different combinations of AgNO_3_ concentrations (% (*w*/*w*)) and pre- and post-sterilization treatment. Bars and error bars are mean g Ag per g grain and SE for three replicates. X-axis indicates the concentration of AgNO_3_ (% (*w*/*w*)) used for sterilizing the grains. Color differentiation of bars and error bars indicates combination of pre- and post-sterilization treatments. Pale red: grains were dry before sterilization and rinsed in ddH_2_O after sterilization, Dark red: grains were soaked for 20 h before sterilization and rinsed in ddH_2_O after sterilization, Pale blue: grains were dry before sterilization and rinsed in ddH_2_O and 1% (*w*/*w*) NaCl after sterilization, Dark blue: grains were soaked for 20 h before sterilization and rinsed in ddH2O and 1% (*w*/*w*) NaCl after sterilization. Different letters between bars denote a statistically significant difference based on a Tukey’s test of least-squares means adjusted for multiple comparisons (22 estimates). The three-way interaction of AgNO_3_ concentration, pre-sterilization treatment, and post-sterilization treatment was statistically significant: P-value: .03. Least-squares means and Tukey’s test results are specified in detail in Appendix A.

**Figure 5 plants-09-00372-f005:**
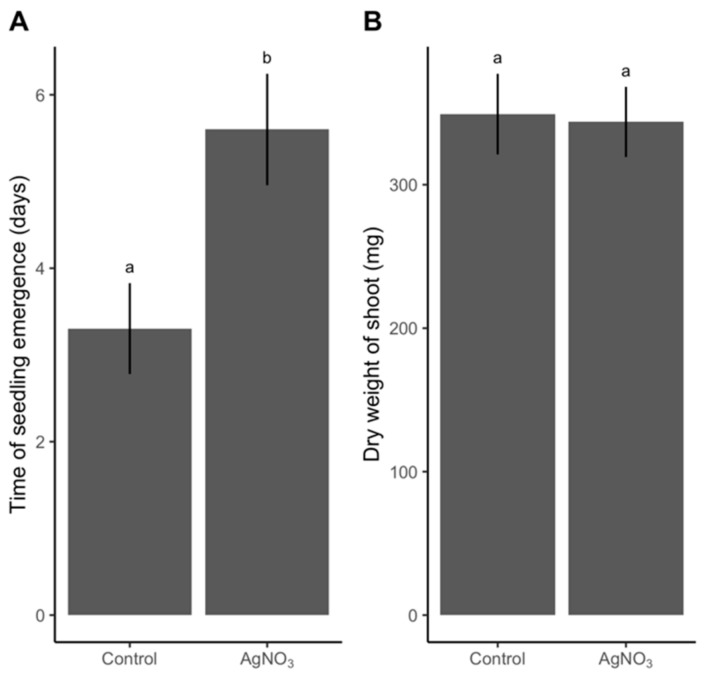
Difference in (**A**) time of seedling emergence and (**B**) shoot dry weight after 25 days between grains of spring barley receiving either no sterilization (Control) or a harsh sterilization: 3% (*w*/*w*) AgNO_3_ (AgNO_3_). n = 23 for the control treatment and n = 20 for the 3% (*w*/*w*) AgNO_3_ treatment. Values are given as mean (SE). Different letters between bars denote a significant difference (P-value <0.05) based on a t-test.

**Figure 6 plants-09-00372-f006:**
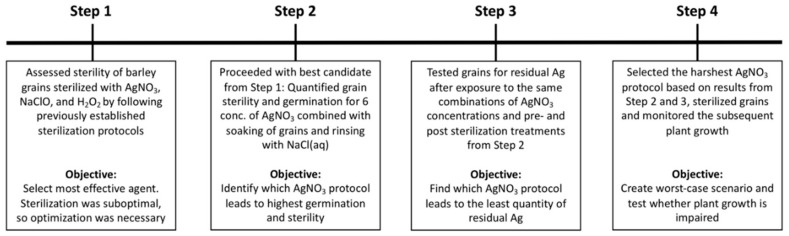
Visual illustration of the four experiments used in this paper. The four separate experiments were necessary to find a protocol for sterilizing barley grains with high sterility and germination, without a lingering sterilization effect and without damage to subsequent plant growth.

**Table 1 plants-09-00372-t001:** Schematic overview of the 22 treatment combinations of Step 2 of the experiment.

Pre-sterilization	AgNO_3_	Post-Sterilization	Media	Pre-Sterilization	AgNO_3_	Post-Sterilization	Media
No soaking (dry grains)	0% (*w*/*w*)	ddH_2_O	PDA	Soaking 20 h	0% (*w*/*w*)	ddH_2_O	PDA
			TSA				TSA
	0.05% (*w*/*w*)	ddH_2_O	PDA		0.05% (*w*/*w*)	ddH_2_O	PDA
			TSA				TSA
		1% NaCl (*w*/*w*) and ddH_2_O	PDA			1% NaCl (*w*/*w*) and ddH_2_O	PDA
			TSA				TSA
	0.2% (*w*/*w*)	ddH_2_O	PDA		0.2% (*w*/*w*)	ddH_2_O	PDA
			TSA				TSA
		1% NaCl (*w*/*w*) and ddH_2_O	PDA			1% NaCl (*w*/*w*) and ddH_2_O	PDA
			TSA				TSA
	1% (*w*/*w*)	ddH_2_O	PDA		1% (*w*/*w*)	ddH_2_O	PDA
			TSA				TSA
		1% NaCl (*w*/*w*) and ddH_2_O	PDA			1% NaCl (*w*/*w*) and ddH_2_O	PDA
			TSA				TSA
	3% (*w*/*w*)	ddH_2_O	PDA		3% (*w*/*w*)	ddH_2_O	PDA
			TSA				TSA
		1% NaCl (*w*/*w*) and ddH_2_O	PDA			1% NaCl (*w*/*w*) and ddH_2_O	PDA
			TSA				TSA
	10% (*w*/*w*)	ddH_2_O	PDA		10% (*w*/*w*)	ddH_2_O	PDA
			TSA				TSA
		1% NaCl (*w*/*w*) and ddH_2_O	PDA			1% NaCl (*w*/*w*) and ddH_2_O	PDA
			TSA				TSA

The two boxes divide grains, which received either no pre-sterilization treatment or were soaked in tap water for 20 h prior to sterilization. Each pre-sterilization treatment was tested against six concentrations of AgNO_3_. For each combination of pre-sterilization treatment and AgNO_3_ (0.05%–10%(*w*/*w*)), grains were either rinsed with or without 1% (*w*/*w*) NaCl. Grains that were not treated with AgNO_3_ (controls) were only rinsed in sterile ddH_2_O. n = 50.

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
