# Peer review of "AgNO3 Sterilizes Grains of Barley (Hordeum vulgare) without Inhibiting Germination—A Necessary Tool for Plant–Microbiome Research"

_plants, 2020, doi:10.3390/plants9030372_

Round 1
Author Response
Dear Reviewer,
Please see the attachment.
Kind regards,
Victoria

Reviewer 2 Report
This study mainly investigated the search of the most efficient protocol to obtain the sterilization of grain, due to the need to remove the microbiome from seeds before the studying of the interaction between plants and microorganisms. As the authors correctly report, for some plants this problem does not exist. For example, for Arabidopsis there is the possibility to use sterile seeds. However, for most plants, sterile seeds are not available. In this context, several sterilization protocols have been evaluated over time, and each laboratory uses the one with which it is most confidential, without having real data to support the efficiency of their protocols. Therefore, even if the topic of the paper is not entirely innovative, I believe it can make a great contribution in order to better standardize routine procedures.
The authors in this paper try different sterilization methodologies, using the most common sterilization agents (NaClO, H2O2, and AgNO3). Through a good experimental design, they gradually exclude the sterilizing agents, finding the combination that can allow a good compromise between sterilization and germination index. Even if the introduction is well written, the authors limited the discussion to the need to have a high yield of sterility without affecting the well-being of the seeds. It would be better to specify the mechanism of action by which the sterilizing agents act.
Minor revision:
Figure 1: the percentage concentrations of each individual sterilizing agent used in the experiment should be added, directly in the figure, or in the correspective caption.
Generally in all the MS there are small errors regarding the units of measurement of the concentrations. It would be appropriate express whether 1% (w / v) or 1% (v / v) for each mixture. An example of this error can be found on line 171, 173, 179, etc. The same problem have to be fixed in all the Figures.
Author Response
Dear Reviewer,
Thank you for your feedback!
Please see the attachment.
Kind regards,
Victoria Munkager

Reviewer 3 Report
Overall Comment.
It was a great pleasure to read this very well-written, well-planned and scientifically valuable study, describing a novel chemical method of seed sterilization using AgNO3, which is then neutralized using NaCl.
There are no major comments. Some minor comments include
There are some inconsistencies in the units (hours or h) and space between number and unit.
Line 35. microorganisms, not microorganism.
Line 104-105. Since Results are placed before Methods, but words such as PDA and TSA are being introduced, suggest the full forms be given here.
Line 172. Consider using concentrations instead of levels, as levels are generally used for liquids that one can see the level of.
Figure 4. Use small h instead of H. one NaC needs to be changed to NaCl. Perhaps use Percentage instead of Pct in x-axis or explain in legend. Also perhaps change hours to h and consistently leave space between number and units
Discussion: Can you be sure that slow growing or viable but not culturable bacteria are not present or have been killed after even the most efficacious AgNO3 statement? To put it another way, would the treatment have any bacteriostatic effect? This needs to be discussed.
Also would the delayed germination imply any epigenetic changes which need to be considered while planning future plant-microbiome experiments?
Author Response

(The authors gave the same response as above.)

Round 2
Reviewer 1 Report
The authors addressed my concerns properly.
However, I dare to suggest a title change, to make it fitting better the article content: "AgNO3 sterilizes barley seeds without inhibiting germination – A necessary tool for plant-microbiome experiments"
In fact, you tested only barley, and it is not possible to assume that it will be the same for all plant seeds. Moreover, to be precise, it is not correct to say "sterilization of the microbiome": one can sterilize a living host from its microbiome, instead. Finally, you let the plants grow for just 25 days after seedling emergence, without testing the whole plant lifecycle (therefore "germination" instead of "growth" is appropriate).
Author Response
Dear Reviewer,
We have not changed the title as you suggested, although we added "Hordeum vulgare" and swapped seed out for grain. The title is now:
AgNO3 sterilizes grains of barley (Hordeum vulgare) without inhibiting germination – A necessary tool for plant-microbiome research
Once again thank you for your feedback!
Kind regards,
Victoria